# Test of the 'glymphatic' hypothesis demonstrates diffusive and aquaporin-4-independent solute transport in rodent brain parenchyma

Alex J Smith[1,2]*, Xiaoming Yao[1,2], James A Dix[1,2], Byung-Ju Jin[1,2], Alan S Verkman[1,2]*

[1]Department of Medicine, University of California, San Francisco, San Francisco, United States; [2]Department of Physiology, University of California, San Francisco, San Francisco, United States

**Abstract** Transport of solutes through brain involves diffusion and convection. The importance of convective flow in the subarachnoid and paravascular spaces has long been recognized; a recently proposed 'glymphatic' clearance mechanism additionally suggests that aquaporin-4 (AQP4) water channels facilitate convective transport through brain parenchyma. Here, the major experimental underpinnings of the glymphatic mechanism were re-examined by measurements of solute movement in mouse brain following intracisternal or intraparenchymal solute injection. We found that: (i) transport of fluorescent dextrans in brain parenchyma depended on dextran size in a manner consistent with diffusive rather than convective transport; (ii) transport of dextrans in the parenchymal extracellular space, measured by 2-photon fluorescence recovery after photobleaching, was not affected just after cardiorespiratory arrest; and (iii) *Aqp4* gene deletion did not impair transport of fluorescent solutes from sub-arachnoid space to brain in mice or rats. Our results do not support the proposed glymphatic mechanism of convective solute transport in brain parenchyma.

DOI: https://doi.org/10.7554/eLife.27679.001

*For correspondence: alex.smith@ucsf.edu (AJS); alan.verkman@ucsf.edu (ASV)

**Competing interests:** The authors declare that no competing interests exist.

## Introduction

Solute transport through the extracellular space (ECS) in brain is of considerable importance for the delivery of nutrients and drugs to brain cells and for the clearance of metabolites, neurotransmitters and toxic macromolecules. The ECS consists of the ventricular system and subarachnoid space containing cerebrospinal fluid, the parenchymal ECS in grey and white matter, and para- and perivascular spaces that surround blood vessels (*Hladky and Barrand, 2014*). The conventional paradigm, which is supported by a large body of experimental evidence, is that solute movement through the narrow, tortuous ECS in brain parenchyma occurs by a diffusive mechanism, whose determinants include solute size and physical properties, and ECS structure and composition (*Syková and Nicholson, 2008*; *Verkman, 2013*).

More recently, Nedergaard and colleagues (*Iliff et al., 2012*; *Jessen et al., 2015*) have proposed that convective, 'glymphatic' flow of cerebrospinal fluid though the ECS from the para-arterial to the para-venous spaces is largely responsible for solute transport in parenchymal ECS. Glymphatic transport was proposed to have broad consequences in normal brain physiology and in the diseased brain, including clearance of β-amyloid in neurodegenerative disease (*Iliff et al., 2012*) and metabolic waste products during sleep (*Xie et al., 2013*), and removal of excess fluid in various forms of brain edema (*Thrane et al., 2014*). It was further proposed that AQP4, a water channel expressed

on astrocyte endfeet, facilitates convective solute transport (*Iliff et al., 2012*). The glymphatic mechanism and a role for AQP4 have been questioned based on apparent inconsistencies with prior data on solute transport in brain ECS (*Hladky and Barrand, 2014*; *Spector et al., 2015*), as well as the uncertain plausibility of convective, pressure-driven fluid flow from para-arterial to para-venous spaces through parenchymal ECS (*Asgari et al., 2016*; *Asgari et al., 2015*; *Jin et al., 2016*; *Smith et al., 2015*).

Here, we re-examined the major experimental findings and predictions that underlie the glymphatic mechanism, including effects of solute size and *Aqp4* gene deletion on solute transport in brain ECS, as well as the immediate effects of cessation of cardiac and respiratory pulsations. The experimental data, together with quantitative modeling, do not support a glymphatic, convective mechanism of solute transport through brain parenchyma, but instead support the classical view of diffusive solute transport.

## Results

### Strongly size-dependent macroscopic solute transport in brain parenchyma

Diffusive transport depends strongly on solute size (hydrodynamic radius, $R_h$), whereas convective transport of uncharged solutes through a porous medium is approximately independent of solute size if the ratio of solute size to pore size is less than ~0.5 (*Dechadilok and Deen, 2009*). To assess the relative importance of diffusion and convection in moving solutes from the subarachnoid space into brain parenchyma, fluorescently labeled, fixable dextrans (10, 70 and 2000 kDa) with different diffusional mobilities ($R_h$ of approximately 2, 5 and 12 nm [*Hadjiev and Amsden, 2015*]) were co-injected into the cisterna magna and their distribution in the brain at 60 min was imaged. All three dextrans were seen at the brain surface and in the paravascular spaces after fixation (*Figure 1A*), confirming that solutes applied to the CSF can circulate around the brain and enter the paravascular spaces. Analysis of the fluorescence intensity of each dextran as a function of distance from the brain surface, where tracer is thought to accumulate in a sub-pial space that is contiguous with the paravascular space (*Hladky and Barrand, 2014*), demonstrated that transport into the brain was strongly size-dependent (*Figure 1B*). Modeling of diffusive transport using experimentally measured diffusion coefficients showed that the relative distance of dextran penetration is consistent with purely diffusive transport into the parenchyma (*Figure 1B* right, *Figure 1—figure supplement 1*). However, absolute penetration distances were not predicted accurately by the model, probably because additional factors, such as solute clearance or barriers between the subarachnoid and subpial spaces, also affect the extent of solute penetration in vivo.

In contrast to the parenchyma, the profile and penetration depth of the different size dextrans in the paravascular spaces surrounding penetrating arterioles at the cortical surface was largely independent of their size (*Figure 1C,D*). However, subtle differences in the distribution of each dextran were seen at higher magnification, as only the smallest (10 kDa) dextran was observed accumulating around cell layers surrounding the vessels (*Figure 1E*), suggesting that transport of large solutes in the paravascular spaces surrounding penetrating arterioles may be limited to the Virchow-Robin spaces in the proximal segment.

To study size-dependent movement of fluorescent solutes deeper in the brain under conditions where solutes are applied directly to the parenchyma, a small, ~1 nL volume of the dextran mixture was injected directly into the striatum of anesthetized mice using a fine glass needle and brains were fixed at 10 or 60 min after injection. At early times points the distribution of all three dextrans was similar (*Figure 2A*), suggesting similar convection into the brain during the pressure pulses applied to the injection pipette. However, dextran penetration from the injection site into the surrounding parenchyma at 60 min was strongly size-dependent (*Figure 2A,B*). Modeling indicated that the measured penetration distance at 60 min was consistent with purely diffusive transport (*Figure 2B*, *Figure 2—figure supplement 1*). Occasionally, when the injected dextran entered the paravascular space of vessels near the injection site, both the large and small dextrans traveled substantial distances away from the injection site (*Figure 2C*). Together, these results suggest that solute movement in brain parenchyma is primarily diffusive and non-directional, and support the long-standing idea

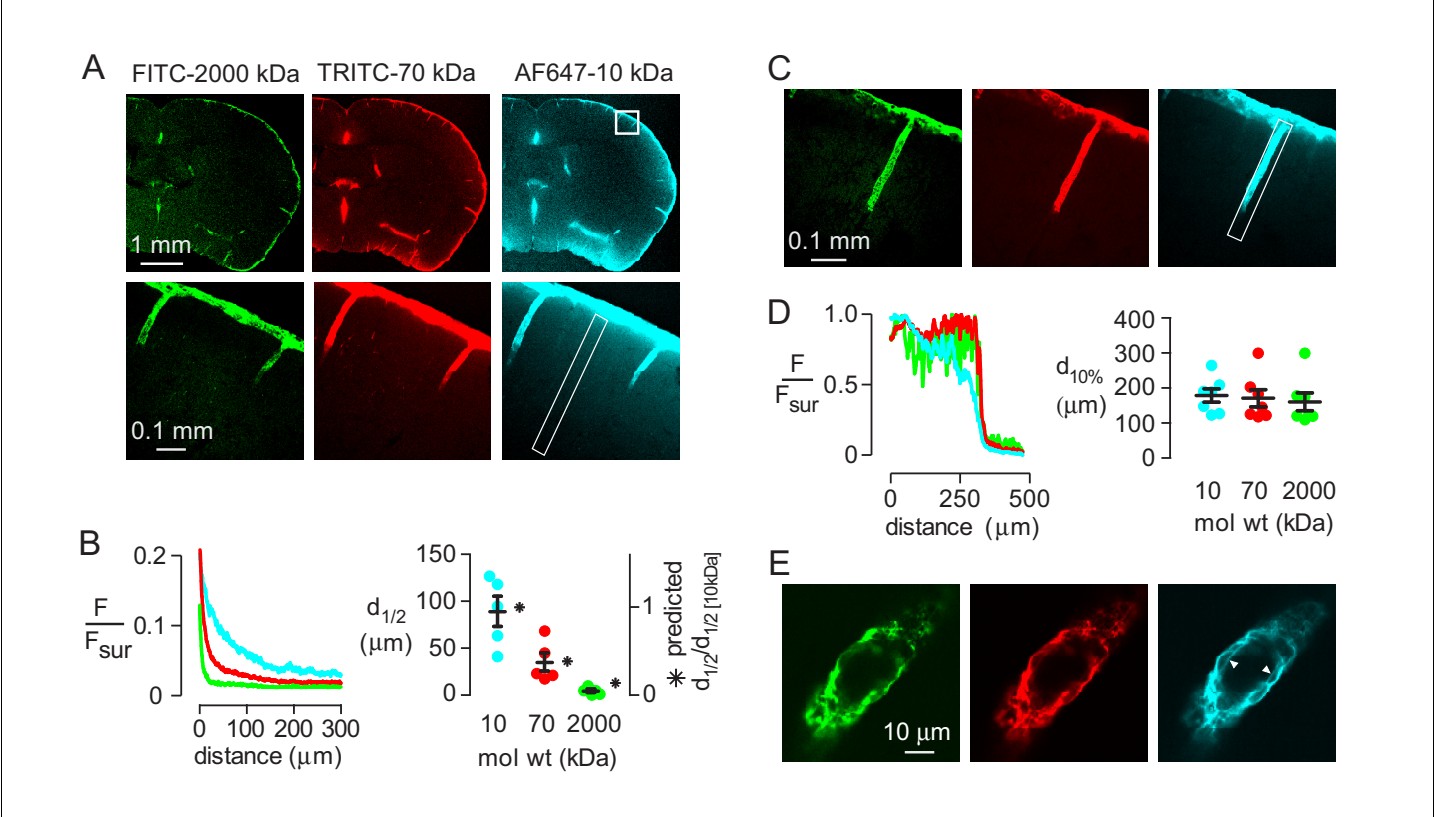

**Figure 1.** Strongly size-dependent penetration of dextrans into mouse brain from the subarachnoid space following intracisternal injection. (A) Confocal images showing the distribution of 10, 70 and 2000 kDa fixable dextrans in brain 60 min after their co-injection into the cisterna magna. The lower panels show a higher magnification view of the boxed region in the top right panel. (B) (left) Profile of dextran fluorescence, normalized to fluorescence at the brain surface, as a function of distance from the cortical surface for the rectangular region shown in the lower right panel in A. (right) Cortical depth at which measured fluorescence intensity decreases to half its value at the cortical surface ($d_{1/2}$, circles, left axis, n = 5 mice, black bars mean ± S.E.M.), and the relative distance that each fluorophore is predicted penetrate in 60 min by diffusion alone (asterisks, right axis). (C) Confocal image of brain cortex 60 min after cisternal injection showing fluorescence along the paravascular space of a penetrating arteriole. (D) (left) Relative fluorescence as a function of distance from the parenchymal surface in the paravascular space outlined on the right of C, measured for each dextran. (right) Distance at which fluorescence intensity decreases to 10% of its initial value (mean ± S.E.M., seven arterioles). (E) High magnification confocal slice through a cortical penetrating arteriole showing subtle differences in the distributions of 10, 70 and 2000 kDa fixable dextrans in the paravascular sheath at 60 min after cisternal injection.

DOI: https://doi.org/10.7554/eLife.27679.002

The following figure supplement is available for figure 1:

**Figure supplement 1.** 1-D diffusion modeling of solute transport from the subarachnoid into the brain parenchyma.

DOI: https://doi.org/10.7554/eLife.27679.003

that paravascular spaces act as a low resistance pathway for convective clearance of solutes from the brain.

## Fluorescence recovery after photobleaching shows non-directional solute transport in brain parenchyma

Direct imaging of solute transfer from the paravascular space to the parenchyma, as reported by (*Iliff et al., 2012*), does not distinguish between vectorial convection and non-vectorial diffusion down a concentration gradient. Fluorescence recovery after photobleaching can detect spatial anisotropy in solute movement due to convection or diffusional anisotropy (*Papadopoulos et al., 2005*; *Sullivan et al., 2009*). We used 2-photon fluorescence recovery after photobleaching through a cranial window to visualize solute movement in brain ECS. Raster scan-type bleaching under these circumstances is predicted to bleach a disk of depth 3 µm (*Mazza et al., 2008*). Initial experiments (*Figure 3—figure supplement 1*) confirmed that that recovery time was approximately linear with

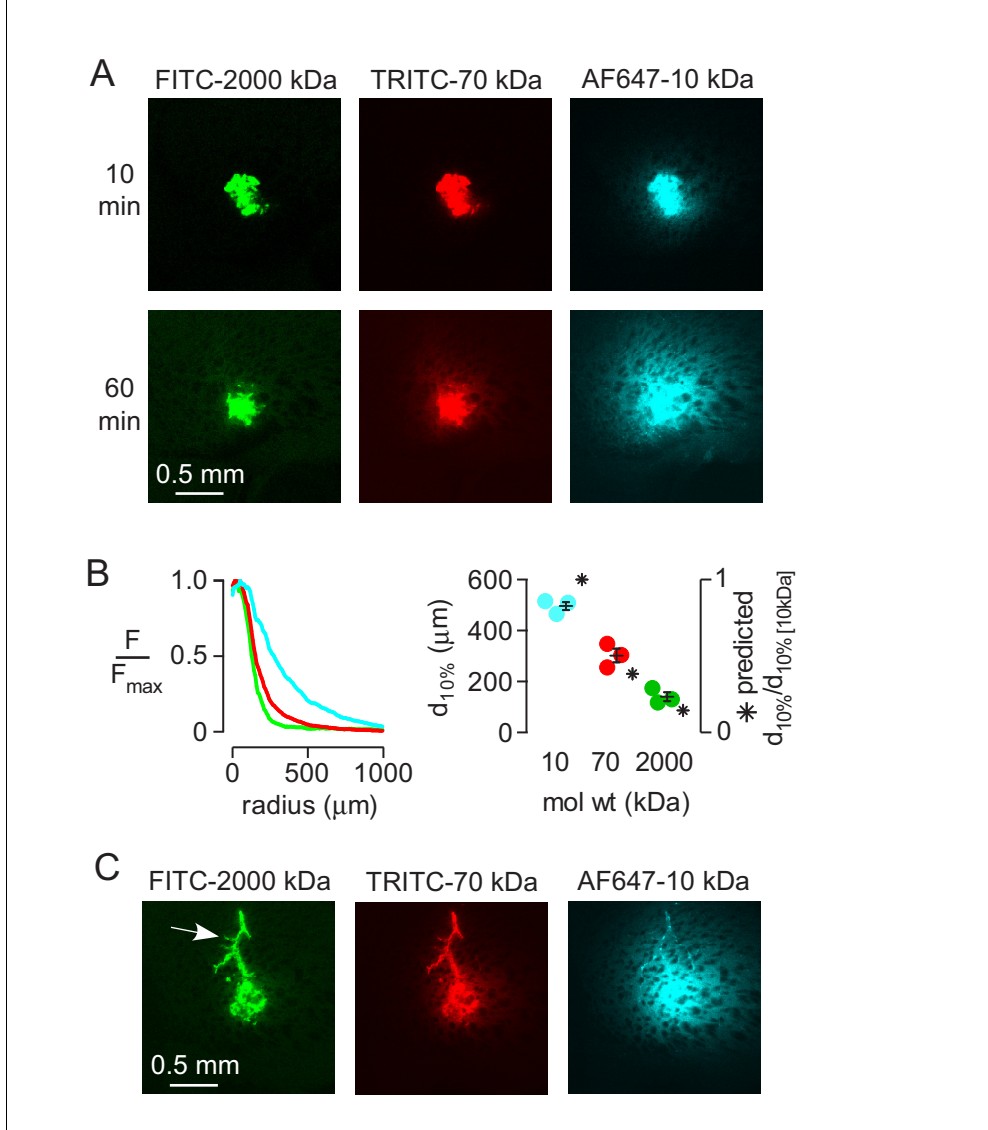

**Figure 2.** Strongly size-dependent transport of dextrans after direct injection into brain parenchyma. (**A**) Confocal images showing the distribution of 10, 70 and 2000 kDa fixable dextrans at 10 and 60 min after coinjection into the striatum. (**B**) (left) Relative fluorescence of each dextran integrated over a circular region centered on the injection site, as a function of radial distance from the center, for the images shown in A. (right) Individual results and mean ± S.E.M. for three experiments showing the distance at which fluorescence decreases to 10% of that at the injection site for each dextran. Asterisks indicate the theoretical relative distance that would be expected for each dextran by diffusion alone. (**C**) Transport of injected dextrans into the paravascular space of vessels running through the injection site occurred independently of size.

DOI: https://doi.org/10.7554/eLife.27679.004

The following figure supplement is available for figure 2:

**Figure supplement 1.** 3-D diffusion modeling of solute transport in the brain cortex.

DOI: https://doi.org/10.7554/eLife.27679.005

---

radius of the bleached region for bleach regions less than ~40 μm diameter, as reported before (*Mazza et al., 2008*), and that recovery time for a 500 kDa dextran in cortical parenchyma was ~3 fold slower than in a watery solution in vitro (*Binder et al., 2004*). In contrast, recovery from bleaching in paravascular regions was substantially faster than that observed in vitro (*Figure 3—figure supplement 2*), supporting a role for convection in this area.

FITC-dextran (500 kDa) was injected into brain cortex (*Figure 3A*, left) and a circular spot of 10 µm diameter was bleached by ~50% (*Figure 3A* right, center). The data were analyzed by several methods designed to detect directional (convective) effects. In one approach, possible anisotropic fluorescence recovery was investigated by dividing the bleached region in quadrants and analyzing recovery in each quadrant, which showed similar recovery rates (*Figure 3B*). Simulations with applied convective flow demonstrated different rates of recovery in each quadrant with convection (*Figure 3—figure supplement 3A*). In a second approach, recovery was analyzed in different regions along the expected direction of convective flow, away from a nearby penetrating arteriole. Comparison of experimental results (*Figure 3C*) with simulated convection supports the absence of convective flow away from the arteriole. Finally, displacement of the centroid of the darkened bleached

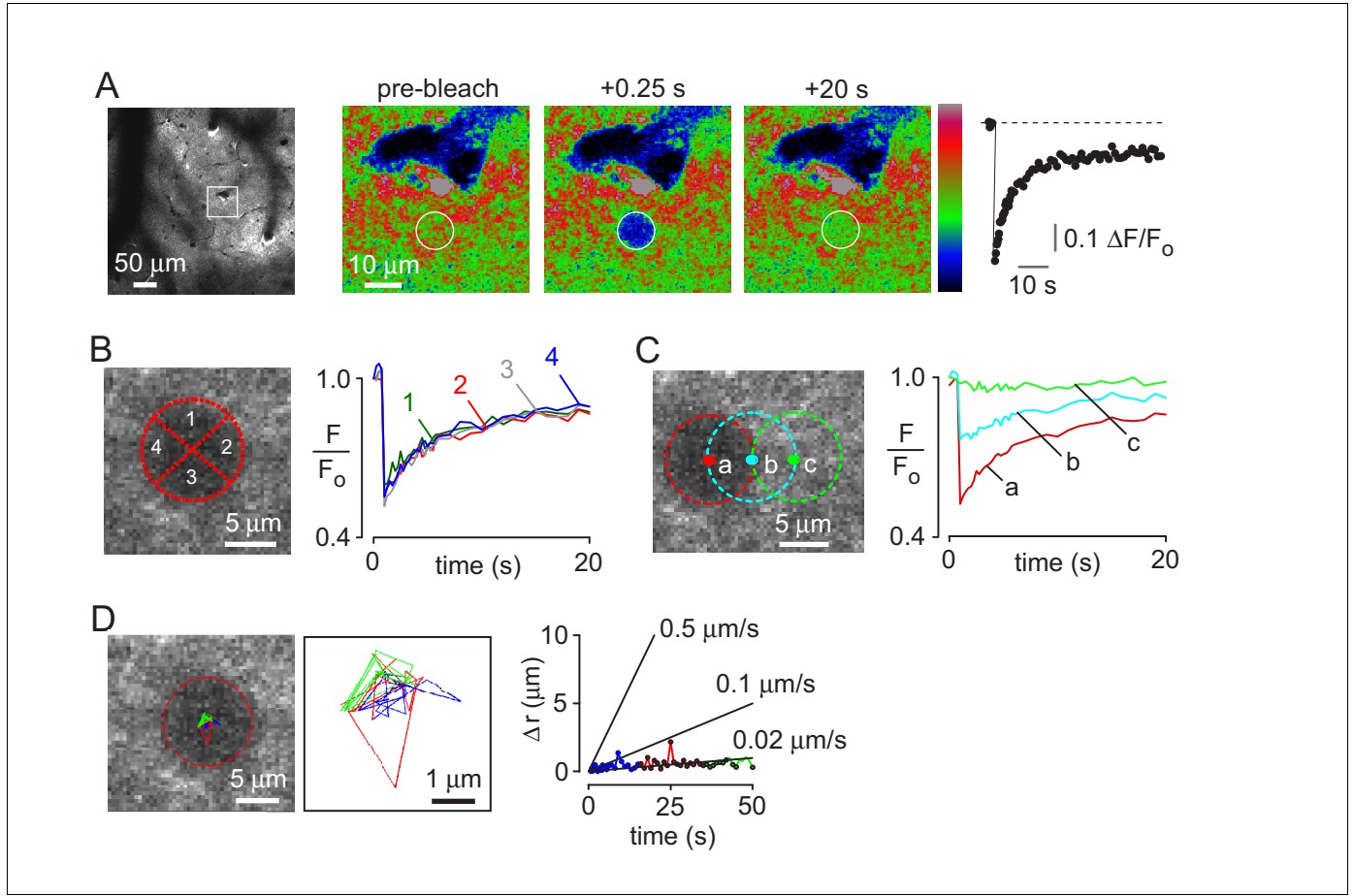

**Figure 3.** Two-photon photobleaching demonstrates non-directional solute transport in brain parenchyma. (**A**) (left panel) Image of 500 kDa FITC-dextran in mouse cortex visualized by 2-photon fluorescence microscopy at a depth of 60 µm. (center panels) Photobleaching of a 10 µm diameter disk immediately adjacent to a penetrating arteriole showing partial recovery of fluorescence in the bleached region 20 s after bleaching. Images are pseudocolored for intensity. (right) Kinetics of fluorescence recovery in the bleached area. (**B**) Kinetics of fluorescence recovery within subdomains 1–4 of the original bleached area, showing spatially homogenous recovery. (**C**) Time course of fluorescence measured at different positions away from a nearby arteriole demonstrates confinement of bleached molecules to the initial bleaching area. (**D**) Positional tracking of the bleached area during recovery (left and center panels), and displacement over time (right panel).

DOI: https://doi.org/10.7554/eLife.27679.006

The following figure supplements are available for figure 3:

**Figure supplement 1.** Characterization of 2-photon photobleaching of 500 kDa FITC-dextran.
DOI: https://doi.org/10.7554/eLife.27679.007

**Figure supplement 2.** Rapid solute transport in the paravascular space.
DOI: https://doi.org/10.7554/eLife.27679.008

**Figure supplement 3.** Quantification of convection verses diffusion from 2-photon photobleaching data.
DOI: https://doi.org/10.7554/eLife.27679.009

spot was determined in each frame after photobleaching, which would occur with convection, as simulated (*Figure 3—figure supplement 3C*). No evidence for directional movement was seen (*Figure 3D*), with an upper limit of ~1 μm/min to convective flow, consistent with previous, macroscopic, failures to detect convection in grey matter (*Rosenberg et al., 1980*).

## Unaltered solute transport in brain just after cardiorespiratory arrest

Iliff *et al.* (*Iliff et al., 2013*) reported ~50% slowing of solute transport from paravascular space into parenchyma following unilateral carotid artery ligation in mice and concluded that arterial pulsations were driving convective transport in the parenchyma. These results are open to alternative interpretations as they do not distinguish between arterial pulsations directly driving convection into the parenchyma, or, alternatively, driving convection in the paravascular spaces followed by diffusion of solutes into the parenchyma. Additionally, chronic partial hypoxia may slow parenchymal transport by reducing ECS transport. To distinguish between these possibilities, photobleaching experiments were done to measure solute transport in the parenchyma following sudden cardiorespiratory arrest in which arterial and respiratory pulsations were abruptly terminated by intravenous injection of succinylcholine followed by air embolism. Acute cardiac arrest by this method does not cause significant swelling of brain cells for several minutes, until anoxic depolarization occurs (*Risher et al., 2009*). FITC-dextran (150 kDa) was used for these experiments and bleached by ~30%, resulting in fairly rapid recovery of fluorescence, enabling repeat bleaching in the same area (*Figure 4A* top panels, *Video 1*). Remarkably, the half-time for fluorescence recovery was unchanged in the first 2–3 min following cardiorespiratory arrest compared with that prior to arrest (*Figure 4A* middle panels, *Video 2*), with recovery slowing thereafter as a consequence of brain swelling produced by anoxic depolarization (*Figure 4A* bottom panels, *Videos 3* and *4*). These results and the associated quantitative analysis (*Figure 4B,C*) confirm that anoxic swelling severely slows extracellular solute transport, and demonstrate that respiratory and cardiac pulsations do not facilitate extracellular solute transport in the parenchyma.

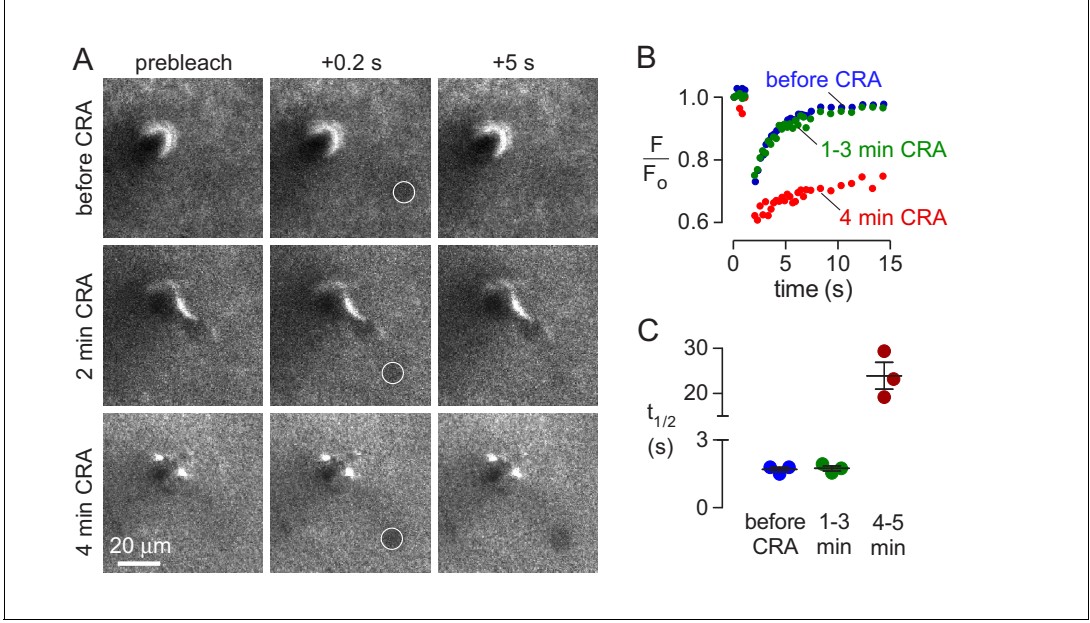

**Figure 4.** Effects of abrupt cessation of cardiac and respiratory pulsations on solute transport in brain parenchyma. (**A**) Individual frames of a 10 μm diameter circular region (white circle) taken prior to (top panels), immediately following (middle panels) and 4 min after (bottom panels) cardiorespiratory arrest. (**B**) Fluorescence recovery curves for experiments as in A showing average of 2–3 trials done before (basal), at 1–3 min, or at 4–5 min following cardiorespiratory arrest. (**C**) Summary of fluorescence recovery half-times ($t_{1/2}$) before and following cardiopulmonary arrest (mean ± S.E.M., 3 mice). Recovery half-time was not significantly different between baseline and 1–3 min. following cardiorespiratory arrest by t-test (p=0.78).
DOI: https://doi.org/10.7554/eLife.27679.010

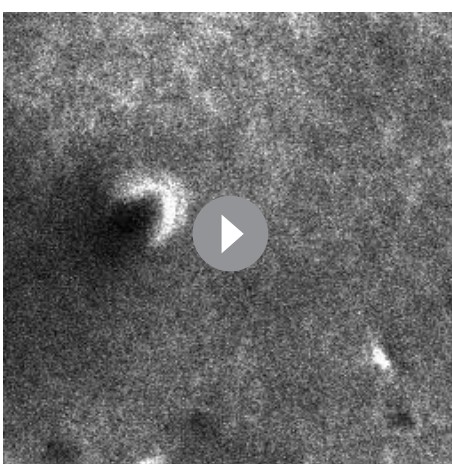

**Video 1.** Photobleaching and fluorescence recovery for 150 kDa FITC-dextran under baseline conditions (corresponds to top panels in *Figure 4A*). Arrow indicates bleached region.
DOI: https://doi.org/10.7554/eLife.27679.011

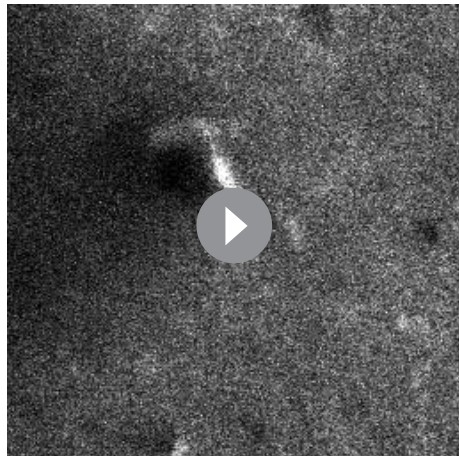

**Video 2.** Photobleaching and fluorescence recovery for 150 kDa FITC-Dextran immediately following cardiorespiratory arrest (note the absence of image jitter caused by heartbeat). Corresponds to middle panels in *Figure 4A*. Arrow indicates bleached region.
DOI: https://doi.org/10.7554/eLife.27679.012

## Aqp4 gene deletion does not reduce solute transport through brain parenchyma

The water channel AQP4 is enriched at astrocyte endfeet in the healthy brain but this polarization is lost in several neurological conditions including in Alzheimer's disease (*Wilcock et al., 2009*) and following stroke (*Steiner et al., 2012*). Iliff *et al.* (*Iliff et al., 2012*) have concluded that perivascular AQP4 facilitates the transfer of solutes from brain CSF to the parenchyma and mediates convective clearance of solutes from the brain (*Iliff et al., 2014*). This conclusion was based largely on experiments that measured the amount of solute transferred from the CSF to the brain parenchyma in $Aqp4^{+/+}$ and $Aqp4^{-/-}$ mice.

We attempted to replicate these experiments by injecting Alexa 647-labelled ovalbumin into the cisterna magna of $Aqp4^{+/+}$ and $Aqp4^{-/-}$ mice, fixing at 30 min after injection, and measuring the fraction of fluorescently stained brain area by intensity-based thresholding as done by Iliff et al (*Iliff et al., 2012*). We found fluorescence at the brain periphery, in the paravascular spaces surrounding vessels, and in parenchyma adjacent to the brain surface (*Figure 5A*), in a similar general pattern to that reported by Iliff et al. However, in contrast to Iliff et al., sections from $Aqp4^{-/-}$ mice showed similar or slightly increased dye penetration than $Aqp4^{+/+}$ mice (*Figure 5A,B*). As there may be regional differences in tracer penetration throughout the brain (*Kress et al., 2014*), we measured fractional tracer penetration in serial coronal sections made at 100 μm intervals through the brain. Again, there was no difference in ovalbumin uptake between $Aqp4^{+/+}$ and $Aqp4^{-/-}$ mice (*Figure 5B*). As the selection of threshold fluorescence is arbitrary, we repeated the analysis at three different threshold levels (*Figure 5C*). Though changing the threshold level altered the fractional fluorescent area, as expected, no significant differences were found in $Aqp4^{+/+}$ vs. $Aqp4^{-/-}$ mice.

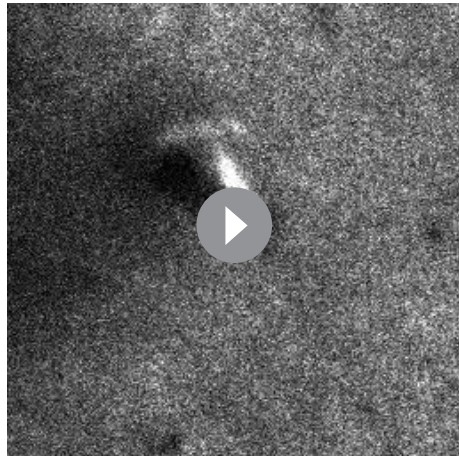

**Video 3.** Photobleaching and fluorescence recovery 3 min after cardiorespiratory arrest and subsequent swelling due to anoxic spreading depolarization. Arrow indicates bleached region.
DOI: https://doi.org/10.7554/eLife.27679.013

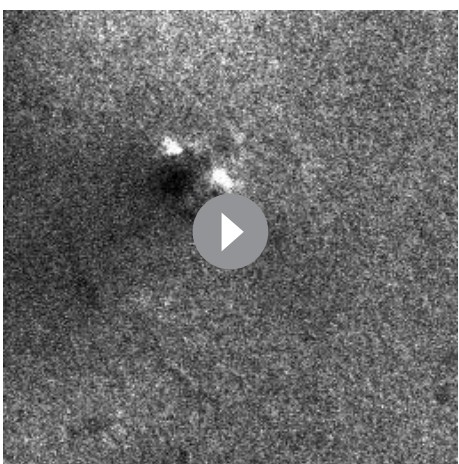

**Video 4.** Photobleaching and fluorescence recovery 4 min after cardiorespiratory arrest demonstrates very slow fluorescence recovery in swollen brain. Corresponds to bottom panels in *Figure 4A*. Arrow indicates bleached region.
DOI: https://doi.org/10.7554/eLife.27679.014

A threshold-based approach for measuring tracer uptake into brain parenchyma does not quantitatively measure the amount of dye entering the parenchyma, as large variations in fluorescence intensity are treated as identical using this approach. As an alternative, we measured intensity of tracer dye in the brain parenchyma relative to that at the brain surface as a function of distance from the cortical surface (*Figure 5D*). Diffusional mobility of large solutes is increased at the cortical surface in $Aqp4^{-/-}$ mice (*Binder et al., 2004*) and extracellular volume is slightly increased (*Yao et al., 2008*; *Zhang and Verkman, 2010*). The combination of these factors results in similar relative movement of solutes into the brain of $Aqp4^{+/+}$ and $Aqp4^{-/-}$ mice in simulations of purely diffusive transport (*Figure 5—figure supplement 1*). In agreement with the simulations, solute diffusion in *Figure 5D* (right) shows similar penetration half-distance of solutes into the brain of $Aqp4^{+/+}$ and $Aqp4^{-/-}$ mice.

As an additional test of the role of AQP4 in solute transport from CSF to brain parenchyma, we performed experiments in recently generated $Aqp4^{-/-}$ rats. The distribution of cisternally injected ovalbumin in rat brain was qualitatively different to that observed in mouse brain (*Figure 5E*, left), with relatively less accumulation at the brain surface and brighter labeling of paravascular spaces in the striatum. Penetration of ovalbumin into the striatal parenchyma from the paravascular space was apparent in both $Aqp4^{+/+}$ and $Aqp4^{-/-}$ rats (*Figure 5E*, center) and quantification of transport of ovalbumin into the parenchyma from the paravascular space showed no significant difference (*Figure 5E*, right). Experiments in both AQP4-deficient mice and rats do not support the conclusion that *Aqp4* deletion impairs transfer of cisternally injected tracers into the brain parenchyma.

The glymphatic system has been proposed to clear toxic peptides from the brain by convective flow through the interstitial space, and failure of this clearance has been proposed to underlie the neuropathology in APP/PS1 mice crossed with *Aqp4* knockout mice (*Xu et al., 2015*). To determine if parenchymal transport of beta-amyloid is directional or affected by *Aqp4* deletion, we coinjected HiLyte647-labelled $A\beta_{1-40}$ peptide and 2000 kDa FITC-dextran into the striatum of $Aqp4^{+/+}$ and $Aqp4^{-/-}$ mice. *Figure 5F* (left) suggests that movement of the Aβpeptide is more complex than movement of dextrans, with a significant immobile fraction and a heterogeneous mobile fraction. Despite the complexities of Aβmetabolism in the brain, which include oligomerization and membrane binding (*Hong et al., 2014*), in addition to degradation and bulk transport (*Tarasoff-Conway et al., 2015*), the distribution of $A\beta_{1-40}$ peptide was generally similar to that observed for injected low molecular mass dextran. Analysis of $A\beta_{1-40}$ distribution, as was done for dextrans in *Figure 2*, indicated a similar $A\beta_{1-40}$ distribution in $Aqp4^{+/+}$ and $Aqp4^{-/-}$ mice (*Figure 5F*, right). These results do not support a significant role for AQP4-facilitated parenchymal convection in the clearance of beta amyloid.

## Discussion

Determining the relative importance of convection versus diffusion in transporting solutes through brain extracellular space is a long-standing problem in neurobiology of particular importance for clearance of toxic protein aggregates (*Hladky and Barrand, 2014*; *Bakker et al., 2016*; *Abbott, 2004*). Some amount of convective flow in the paravascular spaces is generally accepted, although the direction of this flow, its magnitude, and the mechanisms that regulate it remain less clear (*Hladky and Barrand, 2014*). Whereas most previous authors have concluded that solute movement in brain parenchyma is adequately described by simple diffusion, Nedergaard and

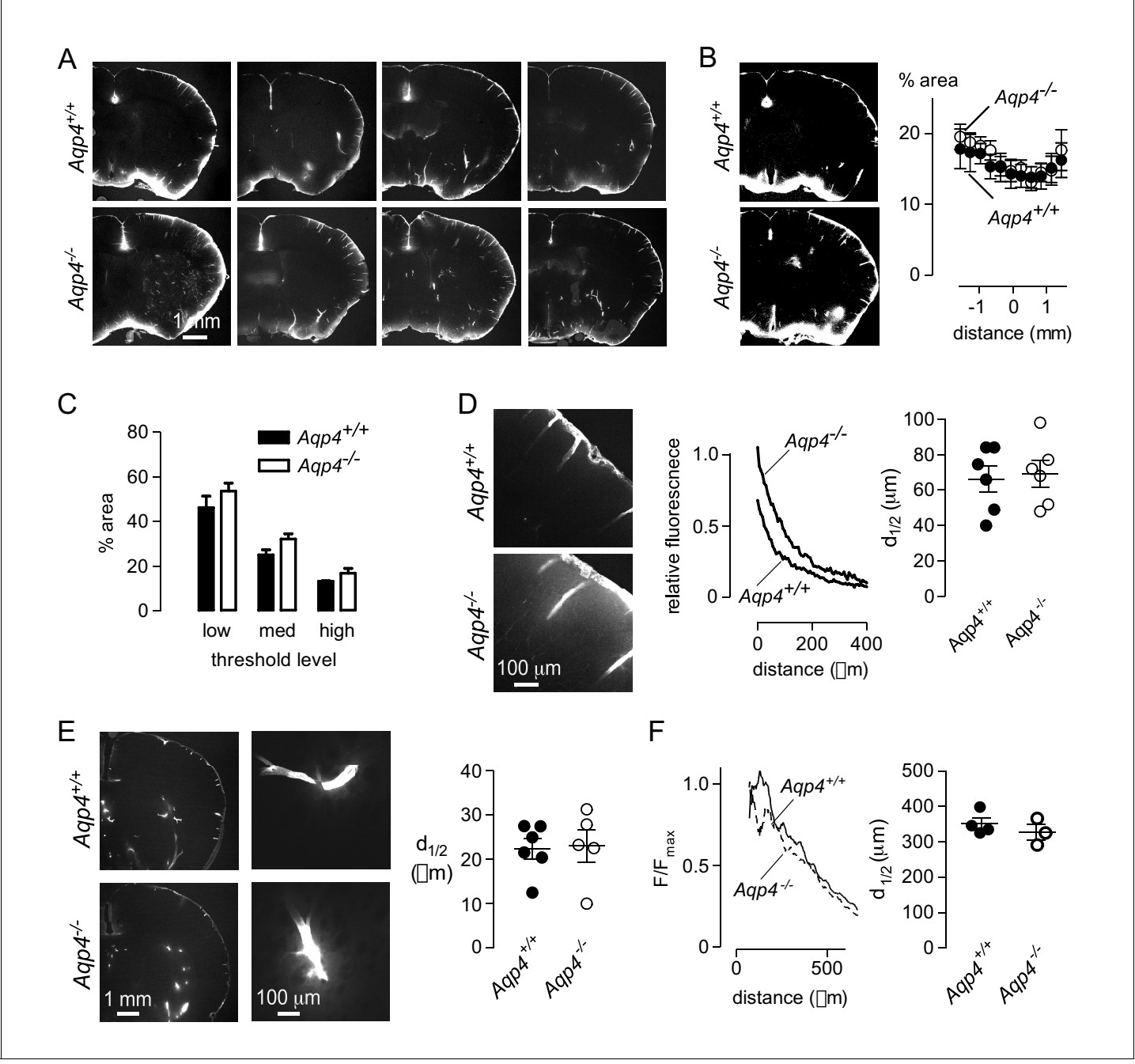

**Figure 5.** *Aqp4* gene deletion does not impair penetration of fluorescent ovalbumin into brain parenchyma from the subarachnoid space. (A) Fluorescence images of brain sections from *Aqp4*$^{+/+}$ and *Aqp4*$^{-/-}$ mice that were fixed 30 min after injection of Alexa647-labelled ovalbumin into the cisterna magna, showing fluorescence in the paravascular spaces and parenchyma near the brain surface. (B) (left) Thresholding approach used to determine the fraction of the section containing labelled ovalbumin. (right) Fractional area of ovalbumin uptake for individual brain sections at indicated distances from the bregma (mean ± S.E.M., six mice per genotype, p=0.72 by two-way ANOVA). (C) Choosing different threshold levels for image analysis altered the area covered by fluorescence but did not reveal genotype-specific differences. (D) (left) Higher magnification images showing penetration of solute from the brain surface into the parenchyma in *Aqp4*$^{+/+}$ and *Aqp4*$^{-/-}$ mice. (center) Fluorescence intensity as a function of distance from the surface, for the sections shown at left. (right) Average half-penetration distance of solute into the parenchyma from the brain surface in slices from six mice per genotype (mean ± S.E.M.). (E) (left) Distribution of Alexa 647-labelled ovalbumin at 30 min after injection into rat brain. (center) Movement of ovalbumin into the parenchyma from the para-arterial spaces. (right) Average half-distance moved by dye from the paravascular spaces into the striatal parenchyma for *Aqp4*$^{+/+}$ (n = 6) and *Aqp4*$^{-/-}$ (n = 5) rats (mean ± S.E.M.). (F) Distribution of Aβ$_{1-40}$ following interparenchymal injection in *Aqp4*$^{+/+}$ and *Aqp4*$^{-/-}$ mice. (Left) Average fluorescence intensity of HiLyte-647 Aβ$_{1-40}$ as a function of radial distance from the injection site. (Right) Distance at which fluorescence decreases to 50% of its value at the center of the injection site (4 *Aqp4*$^{+/+}$ mice; 3 *Aqp4*$^{-/-}$ mice; p=0.42 by t-test).
*Figure 5 continued on next page*

*Figure 5 continued*

DOI: https://doi.org/10.7554/eLife.27679.015

The following figure supplement is available for figure 5:

**Figure supplement 1.** Computation of solute transport in brain parenchyma for different ECS volume fractions.

DOI: https://doi.org/10.7554/eLife.27679.016

colleagues proposed that convective flow in the para-arterial spaces continues into the parenchyma resulting in directional clearance of solutes into the paravenous spaces, by a mechanism requiring AQP4 expression in astrocyte endfeet. Here, we independently tested several major predictions of the glymphatic hypothesis and find that: (i) *Aqp4* deletion does not impair transfer of solutes from CSF into the parenchyma; (ii) movement of fluorescent solutes of different sizes through brain parenchyma is consistent with their diffusion coefficients; and (iii) local movement of solutes in the parenchyma is not impaired just after cardiorespiratory arrest. These results do not support glymphatic, convective solute transport in brain parenchyma.

The experiments reported here add to recent work describing the fate of tracer molecules injected into the brain (*Bedussi et al., 2015*; *Morris et al., 2016*; *Bedussi et al., 2017*). We found size-independent distribution of cisternally injected dextrans into the paravascular spaces, but strongly size-dependent dextran penetration into the parenchyma. These results support convective solute transport in paravascular pathways, but provide evidence against a substantial role for convection in brain parenchyma. An interesting observation was that the smallest (10 kDa) dextran was seen in both the paravascular space between astrocytes and vascular smooth muscle and in the 'perivascular' space between the smooth muscle and endothelium, according to the terminology of Bakker *et al.* (*Bakker et al., 2016*). It remains unclear if accumulation of the small dextran in the perivascular space is due to size-dependent permeability of the living vessel or, alternatively, the early stages of clearance of the low molecular weight dextran along a separate pathway in the peri-arteriolar smooth muscle ring as recently suggested by Morris *et al.* (*Morris et al., 2016*). Direct injection of dextrans into the striatum resulted in their symmetrical, size-dependent movement away from the injection site in a manner consistent with pure diffusion. As with cisternal injections, size-independent accumulation of tracers was observed in the paravascular spaces, consistent with previous measurements of the transport of large solutes during convection-enhanced delivery (*Foley et al., 2012*). The relative homogeneity of the striatum results in approximately isotropic diffusion; however, this is unlikely to be the case in more organized regions of the brain where structural barriers, or regional variation in extracellular space fraction (*McBain et al., 1990*), will likely produce anisotropic diffusion. In white matter tracts, both diffusional anisotropy (*Papadopoulos et al., 2005*) and slow convection (*Rosenberg et al., 1980*) have been demonstrated.

Similar observations were made, with dye visible in both the parenchyma and paravascular areas, when a larger volume of dextran was injected into the cortex and visualized by 2-photon imaging through a cranial window (*Figure 3A*). Recent electron microscopy-level observations of astrocyte fine structure in vitrified samples, where tissue shrinkage during fixation is minimal, demonstrate that the paravascular space may be larger than previously appreciated and that endfoot coverage of vessels is incomplete (*Korogod et al., 2015*). The apparent enrichment of dye around vessels is therefore consistent with a diffusional equilibrium between a fluid-filled paravascular space and a parenchyma with ECS volume fraction of ~0.2. We did not observe directional solute transport in the parenchyma in photobleaching experiments, where recovery of fluorescence in bleached disks was consistently isotropic. Reduction in the rate of solute transfer from the paravascular space to the parenchyma following unilateral internal carotid artery ligation led Iliff *et al.* (*Iliff et al., 2013*) to conclude that cerebral arterial pulsatility is a key driver of paravascular solute influx into and through brain parenchyma. Although the role of arterial pulsation in moving CSF in the subarachnoid and paravascular spaces is well-established (*Rennels et al., 1990*), it had not been previously been suggested that arterial pulsation could drive fluid flow through the interstitial space. We directly assessed the contribution of arterial pulsatility to solute movement in the interstitial space by photobleaching of 150 kDa FITC-dextran and found that solute transport was unaltered in the period immediately following cardiac arrest. The substantial reduction in solute transport following anoxic

depolarization seen at 4 min after cardiorespiratory arrest is consistent with previous reports of astrocyte swelling (*Risher et al., 2009*) and hindered extracellular space diffusion due to increased tortuosity (*Zoremba et al., 2008*). We conclude that the reduction in parenchymal tracer accumulation observed by Iliff *et al.* (*Iliff et al., 2013*) is likely due to factors other than inhibition of pulsatile flow through the interstitial space, such as reduction in ECS volume due to partial ischemia (*Syková et al., 1994*) or inhibition of fluid movement in the paravascular space.

The glymphatic mechanism proposes glial involvement in solute transport facilitated by AQP4; however, we found that *Aqp4* deletion in mice or rats did not reduce, but perhaps slightly increased, the transfer of solutes into brain parenchyma following intracisternal injection. A small increase in parenchymal macromolecular diffusion in $Aqp4^{-/-}$ mice was previously found (*Binder et al., 2004*), which is probably due to an expanded ECS (*Yao et al., 2008*; *Zhang and Verkman, 2010*). The mildly greater solute diffusion in $Aqp4^{-/-}$ than in $Aqp4^{+/+}$ mouse brain could account for the results here. The difference between the results here and the earlier report of Iliff *et al.* (*Iliff et al., 2012*) may be due to differences in experimental methodology or data analysis. For example, beveled glass micropipettes of ~40 μm diameter and pulsed pressure injection were used here, whereas Iliff *et al.* used a 30-gauge metal syringe and constant flow injection for solute application into the cisterna magna. The choice of anesthetics may have also differed between the two sets of experiments (avertin vs. unspecified). The analysis method used by Iliff *et al.*, and reproduced here in *Figure 5B* for comparison purposes, is inherently non-quantitative as it is based on subjective intensity thresholding that treats large differences in fluorescence intensity, indicative of large variation in solute concentration, as equal. Finally, the experiments and their analyses here were done by an investigator blinded to genotype information. Our results also do not support a role for AQP4 in convection-driven clearance of beta amyloid, though they do not rule out the possibility that AQP4 may contribute to the pathology of Alzheimer's disease in other ways.

In conclusion, our results do not support the glymphatic clearance mechanism proposed by Nedergaard and colleagues in which transfer of solutes from cerebrospinal to interstitial fluid requires AQP4-dependent convection in brain parenchyma. Instead, our data are consistent with modeling results demonstrating that parenchymal diffusion, coupled to convective or dispersive flow in the paravascular spaces (*Asgari et al., 2016*; *Jin et al., 2016*), is sufficient to explain movement of solutes in the brain. Further investigation of how AQP4 regulates the structure of the parenchymal and paravascular extracellular spaces is needed to clarify how its deletion or redistribution affects solute clearance from the brain. Our findings suggest the need for re-evaluation of the role AQP4 in other aspects of the glymphatic hypothesis, such as clearance of β-amyloid aggregates from the brain parenchyma and solute clearance during sleep.

## Materials and methods

10 kDa Alexa 647-labelled fixable dextran, 70 kDa tetramethylrhodamine-labeled lysine dextran and 2000 kDa fluorescein-labelled lysine dextran (Life Technologies, Carlsbad, CA) were dissolved at 2.5 mg/ml each in artificial cerebrospinal fluid (aCSF) containing (in mM): 150 NaCl, 3 KCl, 1.4 $CaCl_2$, 0.8 $MgCl_2$, 10 Hepes (pH 7.4) and centrifuged at 13,000 x g for 10 min to remove aggregates. Alexa 647-labelled ovalbumin (Life Technologies) was dissolved at 5 mg/ml in aCSF. HiLyte647-labelled $A\beta_{1-40}$ (Anaspec, Fremont, CA) was initially dissolved at 5 mM in DMSO then diluted to 100 μM in aCSF containing 0.5 mg/ml 200 kDa FITC-dextran, sonicated and spun at 13,000 x g for 10 min prior to injection.

### Animals

$Aqp4^{+/+}$ or $Aqp4^{-/-}$ mice on a CD1 background (*Ma et al., 1997*) of either sex at 3–6 months of age were used for experiments. $Aqp4^{-/-}$ rats were generated using CRISPR-Cas9 technology, which will be reported separately. All experiments were approved by the UCSF Institutional Animal Care and Use Committee (IACUC).

### Brain injections

Mice were anesthetized with 250 mg/kg avertin and mounted on a stereotaxic frame. Body temperature was maintained at 37°C using a heating pad. The skin and muscles overlying the cisterna magna were carefully dissected away to reveal the dura through which a beveled glass micropipette (20 μm

external diameter) was inserted. The pipette shaft was scored at 1 mm intervals to allow estimation of injection volumes, where 1 mm = 0.6 μl for a 0.86 mm internal diameter capillary. For low-volume injections a preliminary measurement was done with each pipette to determine the injected volume as a function of pressure pulse by measurements over 5 min. 10 μl of the dextran solution was pressure injected with a Dagan MPI-100 pressure injector (10 psi, 50–100 ms pulse duration, 1 s duty cycle) into the cisterna magna over 5–7 min. Following injection the pipette was left in place for 5 min to ensure no leakage. The mouse was kept on a heating pad and remained anesthetized until sacrifice by perfusion fixation.

For intraparenchymal injections the scalp was removed and a small burr hole was made in the skull 2.0 mm lateral and 0.5 mm rostral of the bregma. A beveled glass micropipette of ~20 μm external diameter was lowered 3.0 mm into the brain and a small volume (~20 nL) was pressure-injected into the striatum with 5 × 100 ms pulses at 1 Hz and 10 psi. Fixation was performed as described below. For injections in rat brain the same protocol was used as for mouse with the injection volume increased to 30 μl.

## Slice preparation

Mice were transcardially perfused with 50 ml PBS containing 20 units/ml heparin to clear blood and then with 50 ml of 4% paraformaldehyde (PFA) in PBS. The brain was then removed and fixed for a further 24 hr in 4% PFA then rinsed and stored in PBS prior to sectioning. 100 μm thick sequential brain sections were cut on a vibratome (Leica VT1000S) from approximately 2 mm caudal to 2 mm rostral of the bregma, and sections (six per slide) were mounted in ProLong gold antifade reagent (Life Technologies) and covered with a #1 coverslip; antifade was allowed to cure for at least 48 hr prior to imaging.

## Confocal imaging

Sections were imaged on a Nikon C1 confocal microscope using 488, 561 and 639 nm diode lasers for sequential excitation, a triple notch dichroic beam splitter, and 3 detection channels separated by 525/50 nm bandpass, 595/50 nm bandpass and 650 nm long-pass emission filters. Images were captured with 2x/0.06 N.A., 20x/0.5 N.A. or 100x oil/1.4 N.A. objectives.

## Photobleaching measurements

Mice were anesthetized with ketamine/xylazine and a femoral vein catheter fashioned from heat-pulled PE-10 polyethylene tubing was inserted. Mice were then placed in a stereotaxic frame with heating pad and a cranial window of 2–3 mm diameter was made as described (*Holtmaat et al., 2009*). A beveled glass micropipette of ~20 μm diameter containing 25 mg/ml fluorescein-dextran in aCSF was inserted at an angle of 30° into the cortex to a depth of 750 μm and ~200 nl of the dextran solution was pressure injected (10 psi, 50–100 ms pulse duration, 1 s duty cycle) over 3–5 min. The pipette was then withdrawn and a 5 mm diameter #1 glass coverslip was cemented with cyanoacrylate glue over the cranial window. Imaging was done on a Nikon A1R 2-photon scanning microscope using a 25x/1.1 N.A. water immersion objective lens. A 100 × 100 μm field of view at a depth of 60–80 μm from the cortical surface was used for measurements. Excitation was with a Spectra-Physics MaiTai DeepSee IR laser tuned to 840 nm. For measurement of lateral movement of bleached molecules in the parenchyma, a circular spot of 10 μm diameter was bleached with laser intensity tuned to give ~50–60% initial bleach depth, and recovery was followed for 60 s. Bleaching was done before and at different times after induction of cardiorespiratory arrest by iv injection of 100 μl of 5 mg/ml succinylcholine followed by embolization with 1 ml of air 30 s later (*Risher et al., 2009*). Respiratory arrest occurred within 60 s of succinylcholine administration, with cessation of blood flow and the slight movement artifact caused by the pulse. Mean half-recovery times were determined for each condition as well as for in vitro reference experiments done in agarose gels.

## Image analysis

For experiments comparing uptake of tracers into brain parenchyma following injection into the cisterna magna of *Aqp4$^{+/+}$* and *Aqp4$^{-/-}$* mice, images of a whole brain hemisphere were captured with a 2x objective and analyzed as described (*Iliff et al., 2012*), by measuring the fractional of the total brain area above an arbitrary threshold. The investigator performing the sectioning and image

analysis was blinded to genotype information until analysis was complete. For quantitative comparison of labeling intensity as a function of distance from the brain surface, a 20x objective was used to capture three images sequentially from each channel by varying the exposure (1, 4 or 16 µs pixel dwell time) with constant detector gain. The average surface intensity was calculated from the short exposure time images and multiplied by the ratio of exposure durations to allow comparison with parenchymal staining intensity measured at longer exposure times. The linearity of this method was verified by measuring the intensity of a dilution series of each indicator at different exposure times. For determining movement of tracers out of the paravascular space in rat brain a linear analysis region was drawn at right angles to large diameter striatal vessels and fluorescence intensity along this line was determined. The distance at which fluorescence intensity decreased to half its initial value was then determined from four separate vessels for each animal and averaged.

Photobleaching data were analyzed using custom written Matlab algorithms (*Smith et al., 2017*; copy archived at https://github.com/elifesciences-publications/Convection-Diffusion-Analysis-) to track the center of the bleached spot and to determine possible angular asymmetry in fluorescence images during recovery. To identify angular asymmetry the spot was divided into four quadrants (each 90 degrees) and the averaged fluorescence recovery for each segment was plotted. As a more sensitive measurement of convection, the center of the bleached area in each frame during recovery was determined by scanning a circle of 10 µm diameter across the image and determining the position at which the maximum decrease of fluorescence intensity from basal conditions occurred.

## Mathematical modeling

Dye diffusion from the surface to the parenchyma was modeled using the analytical solution to the diffusion equation to compare with experimental data. In vivo diffusion coefficients for dextrans and ovalbumin were taken from the literature as $1.6 \times 10^{-7}$ cm$^2$/s for ovalbumin (*Tao and Nicholson, 1996*), $5.1 \times 10^{-7}$ cm$^2$/s for 10 kDa dextran and $0.75 \times 10^{-7}$ cm$^2$/s for 70 kDa dextran (*Nicholson and Tao, 1993*). The diffusion coefficient for 2000 kDa dextran was estimated as $0.1 \times 10^{-7}$ cm$^2$/s from the value measured in aqueous solution (*Sandrin et al., 2016*) and an assumed tortuosity of 2.64. The effect of increased ECS volume fraction (from $\alpha = 0.2$ to 0.24) for diffusional movement in brain ECS was computationally modeled as described (*Jin et al., 2016*).

## Statistical analysis

Statistical analysis and graphing were performed with Prism 5.0 (GraphPad Software). Sample size was calculated *a priori* using G*Power 3.1 (*Faul et al., 2009*) Sample size was sufficient to detect significant p<0.05) differences with power of 0.8. For experiments in *Figure 4C*, an effect size of 2.5 was predicted on the assumption of a 25% increase in recovery rate with a standard deviation of 10% in the measurement of recovery rate. For mouse experiments in *Figure 5*, an effect size of 2.4 was predicted by estimating the mean and standard deviation of data in *Figure 4F* of ref. (*Iliff et al., 2012*).

# Additional information

### Funding

| Funder | Grant reference number | Author |
|---|---|---|
| National Institutes of Health | EB00415 | Alan S Verkman |
| Guthy-Jackson Charitable Foundation | | Alan S Verkman |
| National Institutes of Health | EY13574 | Alan S Verkman |
| National Institutes of Health | DK72517 | Alan S Verkman |
| National Institutes of Health | DK35124 | Alan S Verkman |
| National Institutes of Health | DK101273 | Alan S Verkman |

The funders had no role in study design, data collection and interpretation, or the decision to submit the work for publication.

## Author contributions

Alex J Smith, Conceptualization, Data curation, Formal analysis, Investigation, Methodology, Writing—original draft, Writing—review and editing; Xiaoming Yao, James A Dix, Conceptualization, Investigation, Methodology; Byung-Ju Jin, Conceptualization, Software, Investigation, Methodology; Alan S Verkman, Conceptualization, Supervision, Funding acquisition, Writing—original draft, Writing—review and editing

## Author ORCIDs

Alex J Smith, http://orcid.org/0000-0002-3034-9137

## Ethics

Animal experimentation: This study was performed in strict accordance with the recommendations in the Guide for the Care and Use of Laboratory Animals of the National Institutes of Health. All of the animals were handled according to approved institutional animal care and use committee (IACUC) protocol #AN108511 of the University of California San Francisco. All surgery was performed under avertin or anesthesia, and every effort was made to minimize suffering.

## Decision letter and Author response

Decision letter https://doi.org/10.7554/eLife.27679.018
Author response https://doi.org/10.7554/eLife.27679.019

# Additional files

## Supplementary files

• Transparent reporting form
DOI: https://doi.org/10.7554/eLife.27679.017

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
