## [Decision Letter]

Thank you for submitting your article "Test of the 'glymphatic' hypothesis demonstrates diffusive and aquaporin-4-independent solute transport in rodent brain parenchyma" for consideration by *eLife*. Your article has been reviewed by three peer reviewers, and the evaluation has been overseen by a Reviewing Editor and Richard Aldrich as the Senior Editor. One of the three reviewers, Devin Binder, has agreed to reveal his identity.

The reviewers have discussed the reviews with one another and the Reviewing Editor has drafted this decision to help you prepare a revised submission.

Summary:

This manuscript reports multiple lines of evidence that solute transport through the brain is independent of APQ4 function, and occurs in large part by diffusion rather than by convection. The results challenge a recently proposed glymphatic mechanism of convection solute transport in brain parenchyma. Experiments directly test and fail to replicate results presented previously by others. In addition, the authors present several other types of compelling evidence to support their conclusions. While overall supporting more of the diffusion mechanism than a "glymphatic" mechanism, this paper provides an incomplete test of the glymphatic mechanism described in Iliff et al. 2012 insofar as no direct test of the paravascular pathway is provided for protein solutes previously hypothesized to follow this pathway for clearance, such as amyloid-β. Tracking fluorescently labeled amyloid-β in the brain to directly refute or confirm Iliff et al. 2012's findings, would greatly strengthen the results. Otherwise, the experiments have been rigorously conducted, properly controlled, properly evaluated and convincing. These results are important for, and should be of interest to, a broad audience. The topic is an important one, and this study highlights the need for further work.

Essential revisions:

1) The clever experiments of Figure 4 demonstrate that the absence of respiratory and cardiac pulsations does not influence intraparenchymal solute diffusion; however, this does not directly test whether there is unchanged solute transport from paravascular space into parenchyma (which would be the analogous experiment to test the hypothesis of Iliff et al. (Iliff et al., 2014)). The authors should repeat this experiment with paravascular to parenchymal measurement over the same time windows (before CRA, 1-3min CRA and 4min CRA) to fully test this hypothesis.

2) The first sentence of the Results states that convective transport is approximately independent of solute size, whereas diffusive transport depends strongly on it. This statement critically underpins the interpretation of the subsequent results that the transport observed is dependent on solute size, and for this reason it should be supported by an authoritative reference. Most readers, myself included, will not be familiar with this topic and it would be important to provide quick access to a reference.

3) In both the current study Smith et al., and the previous study by Iliff et al., 2012, results from AQP4^-/-^ mice play critical roles in the conclusions drawn. Since the conclusions of the two papers are opposite, it is important to compare the experiments conducted in each study. Smith et al. show in their Figure 5 a direct replication attempt of the data shown in Iliff et al. Figure 3. The Smith et data do not replicate that by Iliff et al., but instead suggest no effect of AQP4^-/-^. As noted in the summary, an experiment testing whether protein solutes, preferably A-β, diffuse similarly to dextrans, should be provided. Furthermore, this experiment appears to be underpowered, with only a small number of mice tested, and with significant variability between mice.

[Editors' note: further revisions were requested prior to acceptance, as described below.]

Thank you for resubmitting your work entitled "Test of the 'glymphatic' hypothesis demonstrates diffusive and aquaporin-4-independent solute transport in rodent brain parenchyma" for further consideration at *eLife*. Your revised article has been evaluated by Richard Aldrich (Senior Editor), a Reviewing Editor, and three reviewers.

The manuscript has been improved but there are some remaining issues that need to be addressed before acceptance, as outlined below:

In particular, the reviewers would like further clarification regarding replication numbers. For the critical experiments depicted in Figure panels 5A-D, please provide details regarding numbers of animals supporting each result. In addition, the number of rats used as the species replication experiment is definitely on the low side. Please revise accordingly, and the editors will make a decision, possibly in consultation with reviewers.

---

## [Author Response]

*1) The clever experiments of Figure 4 demonstrate that the absence of respiratory and cardiac pulsations does not influence intraparenchymal solute diffusion; however, this does not directly test whether there is unchanged solute transport from paravascular space into parenchyma (which would be the analogous experiment to test the hypothesis of Iliff et al. (Iliff et al., 2014)). The authors should repeat this experiment with paravascular to parenchymal measurement over the same time windows (before CRA, 1-3min CRA and 4min CRA) to fully test this hypothesis.*

We had initially considered such an experiment but concluded that it is not possible to make measurements of paravascular to parenchymal transport, as done in Iliff et al., 2014, in the time frame of our experiment. The experiment of Iliff et al., 2014 measured the rate of accumulation of dyes in the parenchyma after injection into the cisterna magna and subsequent transport through the perivascular spaces, a process that takes 20-30 min. The time frame for measuring the effect of acute cardiorespiratory arrest on solute transport, prior to anoxic depolarization and tissue swelling, is 3-4 min, meaning that transfer cannot be measured before swelling occurs. The simulations in the supplement to Figure 3 demonstrate that our photobleaching experiments can detect directional convective flow from the paravascular space into the parenchyma. Additional discussion of the differences between our experiment and that of Iliff et al., 2014 has been added (Discussion) to clarify the differences in experimental protocol and the conclusions that can be drawn from them.

*2) The first sentence of the Results states that convective transport is approximately independent of solute size, whereas diffusive transport depends strongly on it. This statement critically underpins the interpretation of the subsequent results that the transport observed is dependent on solute size, and for this reason it should be supported by an authoritative reference. Most readers, myself included, will not be familiar with this topic and it would be important to provide quick access to a reference.*

Additional explanation with reference has been added at the beginning of the Results section.

*3) In both the current study Smith et al., and the previous study by Iliff et al., 2012, results from AQP4^-/-^ mice play critical roles in the conclusions drawn. Since the conclusions of the two papers are opposite, it is important to compare the experiments conducted in each study. Smith et al. show in their Figure 5 a direct replication attempt of the data shown in Iliff et al. Figure 3. The Smith et data do not replicate that by Iliff et al., but instead suggest no effect of AQP4^-/-^. As noted in the summary, an experiment testing whether protein solutes, preferably A-β, diffuse similarly to dextrans, should be provided. Furthermore, this experiment appears to be underpowered, with only a small number of mice tested, and with significant variability between mice.*

Good suggestion. Experiments are now included on the transport of Aβ_1-40_ after striatal injection (in a new panel, Figure 5), which support our conclusions. Also, as requested, we have increased the n for the experiments in Figure 5.

[Editors' note: further revisions were requested prior to acceptance, as described below.]

*In particular, the reviewers would like further clarification regarding replication numbers. For the critical experiments depicted in Figure panels 5A-D, please provide details regarding numbers of animals supporting each result. In addition, the number of rats used as the species replication experiment is definitely on the low side. Please revise accordingly, and the editors will make a decision, possibly in consultation with reviewers.*

In our revised manuscript, the text has been edited to make it clear that 6 *Aqp4*^+/+^ and 6 *Aqp4*^-/-^ mice were used for experiments in Figure 5. We have also performed additional experiments in rats and now report results from 6 *Aqp4*^+/+^ and 5 *Aqp4*^-/-^ rats in Figure 5.